# Driving Factors of Decoupling between Economic Development and Water Consumption in Food and Energy in North-West China—Based on the Tapio-LMDI Method

**Ziyu Pan [1], Zhou Fang [1,\*], Junyu Chen [2,3,\*], Jun Hong [4], Yisong Xu [2] and Shiliang Yang [1]**

[1] Institute of Management Science, Hohai University, Nanjing 210098, China; panziyu@hhu.edu.cn (Z.P.); max388@126.com (S.Y.)

[2] School of Business, Suzhou University of Science and Technology, Suzhou 215009, China; xysszdq@163.com

[3] College of Management and Economics, Tianjin University, Tianjin 300072, China

[4] Economic Management Institute, Suzhou Polytechnic Institute of Agriculture, Suzhou 215008, China; 171312060009@hhu.edu.cn

\* Correspondence: fangzhou@hhu.edu.cn (Z.F.); jychen@usts.edu.cn (J.C.)

**Abstract:** The shortage of water resources has become a notable bottleneck, restricting the economic development of many countries and areas around the word, especially that of North-west China. The Inner Mongolia Autonomous Region and Shaanxi Province are important energy bases and food production areas in North-west China. However, the region is suffering from perennial drought and water shortage, which has become the most significant shortcoming for energy and food production. Guiding the decoupling between regional economic development and water consumption is a critical way to achieve sustainable development. Based on the analysis of the food and energy production value and their water consumption in North-west China from 2009 to 2019, this paper uses the Tapio model to analyze the decoupling relationship between food, energy production, and water consumption, and uses the Logarithmic Mean Divisional Index (LMDI) model to analyze the driving factors affecting decoupling. The results show that most water consumption for food and energy production in North-west China is out of the ideal strong decoupling, the decoupling status is unstable, and re-coupling occurs frequently. The increase in water intensity and the change in industrial structure are the promoting factors of decoupling between production value and water consumption in food and energy in North-west China, while the increase in production value and population size are the main restraining factors. Therefore, in pursuit of strong decoupling, the government should guide the food and energy industry to move toward implementing in water saving measures through policies and promote the enthusiasm and efficiency of the labor force through financial support and other ways. Moreover, ecological protective measures are needed to be strengthened, such as water source protection, and sewage treatment.

**Keywords:** water consumption; decoupling; driving factors; tapio model; LMDI model

## 1. Introduction

Water, energy, and food are the material basis for human society and economic development [1]. The 2011 *Global Risk Report* notes the influence of the complicated relationship among water, energy, and food on regional sustainable development, and suggests that water resources are the most essential resources. Only balanced supply and demand of water resources in industries can ensure the stable supply of food and energy to realize the sustainable development of the economy and society [2]. With the increasing demand for water resources, the contradiction between supply and demand of water resources is becoming increasingly prominent, and the sustainability of the economy and society is also facing significant challenges [3].

"Decoupling" is a basic theory put forward by the Organization for Economic Cooperation and Development (OECD), which explains the connection and unsynchronized

changes between economic growth and resource consumption. Economic development and resource consumption show different trends, corresponding to different decoupling states. In general, if a region achieves economic growth while reducing resource consumption, then the region has an ideal decoupling situation, which is also the goal of decoupling research. In the 1960s, Carter [4] put forward the "decoupling" theory to measure the relationship between economic development and resource consumption. Cesano [5] also pointed out the importance of reducing water consumption for sustainable development from the global economic perspective. Decoupling theory has been widely used by many organizations, including the European Commission [6], and the United Nations Environment Program [7]. Juknys [8] established a decoupling model to study the environmental pollution decoupling situation and proposed two dimensions of primary decoupling and secondary decoupling. Vehmas et al. [9] and Tapio [10] improved and refined the decoupling model, and established a relatively complete decoupling evaluation framework by which the decoupling relationship can be divided into three classifications: decoupling, coupling, and negative decoupling. They further took the percentage of resources increase and decrease caused by economic change and divided the classifications into eight statuses (Table 1). Among them, the strong decoupling status is an ideal state for the sustainable development of resource consumption and economic growth as economic growth occurs, while resource consumption declines; on the other hand, strong negative decoupling is the most unsatisfactory situation in which water consumption increases as economic recession occurs.

**Table 1.** Classification and meaning of decoupling status.

| Classifications | Status | Meanings |
|---|---|---|
| Decoupling | Strong decoupling<br>Weak decoupling<br>Recessive decoupling | Economy grows while resource consumption decreases<br>Increasing rate of resource consumption is less than economy growth rate<br>Reduction rate of resource consumption is greater than economy recession rate |
| Coupling | Expansive coupling<br>Recessive coupling | Increasing rate of resource consumption is equal to economy growth rate<br>Reduction rate of resource consumption is equal to economy recession rate |
| Negative decoupling | Expansive negative decoupling<br>Weak negative decoupling<br>Strong negative decoupling | Increasing rate of resource consumption is greater than economic growth rate<br>Reduction rate of resource consumption is less than economy recession rate<br>Economy decreases while resource consumption increases |

The current empirical research on decoupling aims to pursue strong decoupling and avoid strong negative decoupling. Femia and Luks [11] established a policy framework for the overall strategy of decoupling ecological sustainability from individual well-being at the application level, pointing out the limitations of traditional environmental policies from the perspective of sustainability. Most scholars have studied the decoupling between economic growth and resource consumption at the national and local levels: Tasbasi [12] used the WPAT equation to measure the effects of population, income, and intensity on water demand in the Balkans. Yousaf [13] and Climation and Pardo [14] studied the decoupling between economic growth and resource consumption from national or regional perspectives. Most Chinese scholar's research focuses on the unsynchronized changes of macroeconomic growth and resource consumption in the study of decoupling theory. In terms of land use, Shao and Ge [15] and Zhu et al. [16] introduced the decoupling theory to evaluate the decoupling state between an urban low-carbon economy and intensive land use, and put forward land management policy suggestions for the realization of sustainable urban development. In the field of water resources utilization. Pan and Chen [17] constructed a decoupling evaluation model and proposed regulation strategies for promoting the coordinated development of water resource utilization and the economy. In the field of energy consumption, Lu and Mao [18] deduced a method to evaluate the decoupling of environmental load in the process of economic growth based on the IPAT model and suggested an energy policy for how to cross the "environmental mountain". In terms of carbon emissions, Wang and Yang [19] and Zhou et al. [20] constructed an

expanded decomposition model for decoupling elasticity and the effort index of industrial carbon emissions. The initiative provides a theoretical basis for the development of emission reduction strategies and policies. In general, decoupling research objectively reflects the nature of the unsynchronized change between economic growth and material consumption and has reference value for guiding regional economic growth and realizing resource allocation. However, most of the current decoupling studies are based on macroeconomic conditions, and there are few analyses on water decoupling for related industries, especially from the perspective of ''water–energy–food''. Energy and food are interrelated products, and their production processes are inseparable from the supply of water resources [1]. Current proposals have usually overlooked the ties between resources, and the decoupling situation cannot be reflected if the relationships between economic growth and water consumption of energy and food were not studied from the perspective of water–energy–food. Moreover, the existing research on water resource decoupling mainly focuses on finding the decoupling status, and there are few study on detailed decomposition of decoupling status and that explore its driving factors, making it difficult to provide targeted suggestions in decoupling. Therefore, the decoupling theory creates an opportunity to better understand the relationship between economic development and resource consumption. It is feasible to determine the decoupling relationship and develop targeted solutions for sustainable development.

This research combines the analysis of decoupling status with the decomposition of decoupling driving factors, which is implemented in the two-stage Tapio-LMDI method. By combining the two models, we can put forward more targeted policy suggestions from the perspective of the influencing factors of decoupling on the basis of identifying the decoupling status. Specifically, first, the CROPWAT model, which was developed by the International Food and Agriculture Organization (FAO, Rome, Italy), is used to improve the water consumption measurement in this study. Compared with the common quota method (area ratio × total water consumption = water consumption) in the existing literature [21], this model considers environmental factors (soil, climate) and measures water consumption more accurately through biophysical processes. It is necessary to measure food–water consumption accurately as it can provide strong support for precise evaluation of the decoupling relationship between food and water. Second, as water, energy, and food are interrelated resources, changes in one or more variables will be transmitted to others, which makes the changes of all variables to have evolutionary characteristics. Compared with other methods for evaluating decoupling [12,18] which have limitations of incomplete classification and insufficient quantitative information, the Tapio model can finely measure the decoupling degree [22], which provides a certain advantage of judging the evolution process of the decoupling status. Third, considering the fact that the Logarithmic Mean Divisional Index (LMDI) can decompose all factors without residual error [23,24], it is used to quantitatively analyze the influence of variables on resource consumption [25–27].

China is troubled by the trade-off between economic development and resource consumption, especially in North-west China. The Inner Mongolia Autonomous Region and Shaanxi Province are important energy bases and key areas of food production in North-west China, which play important roles in ensuring China's energy and food security [28]. With the growth of energy and food demand in North-west China, water resources have become a rigid constraint restricting the development of energy and food industries in this region [29,30]. Only by clarifying the decoupling relationship between energy and food production value and water consumption and by identifying the driving factors, can we find the mechanism of optimizing resource allocation and alleviate the harm of energy and food water crises to the economy and society. This is of great theoretical and practical significance to ensure the sustainable development of "water–energy–food" in North-west China. Therefore, in this research, the Tapio model is used to evaluate the decoupling status between economy and water resources of food and energy industry in North-west China, and the LMDI model is used to analyze the driving effect on water consumption variation, and to reduce water resource consumption, as this is helpful in achieving strong

decoupling. The driving factors of decoupling between energy and food production value and water consumption in North-west China are studied, providing a decision-making basis for local governments to ensure regional sustainable development ability and to provide a reference for solving global sustainable development problems.

## 2. Materials and Methods

### 2.1. Study Area

Richness in coal and shortage of water are typical characteristics of the Inner Mongolia Autonomous Region and the Shaanxi Province (Figure 1). By referring to the 2020 *China Statistical Yearbook* [31], the 2020 *Inner Mongolia Statistical Yearbook* [32], and the 2020 *Shaanxi Statistical Yearbook* [33], it can be observed that the sum of the two province's administrative area is 1,388,600 km$^2$, accounting for 14% of China's land area. The sum of the two provinces' population is 64.16 million, accounting for 5% of China's population. The per capita GDP of the Inner Mongolia Autonomous Region and the Shaanxi Province reaches 98% and 96% of the national average, respectively. The available water resources of the two provinces only reach 48% and 53% of the national average level, respectively, while they produce 57% of coal and 7% of the food crops in China, respectively. The water consumption per unit of food production in the Inner Mongolia Autonomous Region and Shaanxi Province was 1.7 times and 1.2 times of the national average, respectively. This means that food water inefficiency in North-west China is greater than in most provinces of the country. This reflects the conflict between the production and the use of water resources in this region, as water shortage is the bottleneck of regional development.

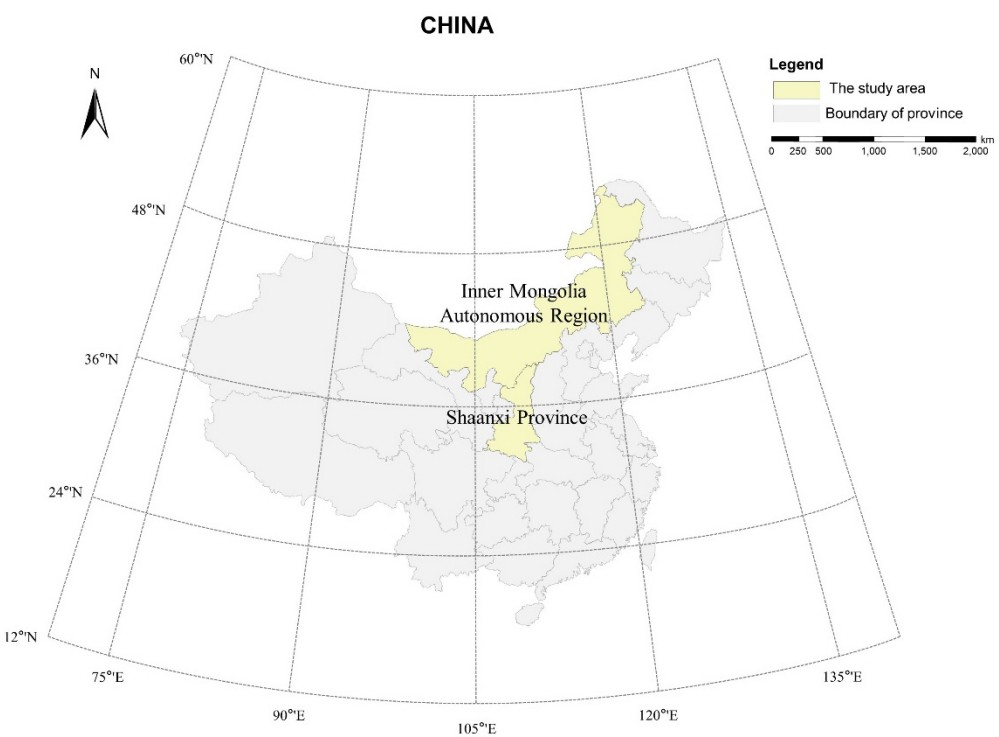

**Figure 1.** The location of the Inner Mongolia Autonomous Region and Shaanxi Province in China.

### 2.2. Analysis Framework of Driving Factors of Decoupling between Economic Development and Water Consumption in Food and Energy

The analysis framework of driving factors of decoupling between economic development and water consumption in food and energy includes four stages (Figure 2): (1) data preparation: food–water consumption was calculated by CROPWAT; energy–water consumption which means water consumption to produce energy, energy production values, food production values, and population were obtained from the *Inner Mongolia Statistical Yearbooks* and the *Shaanxi Statistical Yearbooks*; (2) model calculation: Tapio and

LMDI models were used for index measurement; (3) results and analysis: analysis of decoupling status, driving effect on water consumption variation, and driving factors for decoupling; and (4) suggestions: the measures that are helpful for water decoupling are put forward. The specific steps are shown in the following sections.

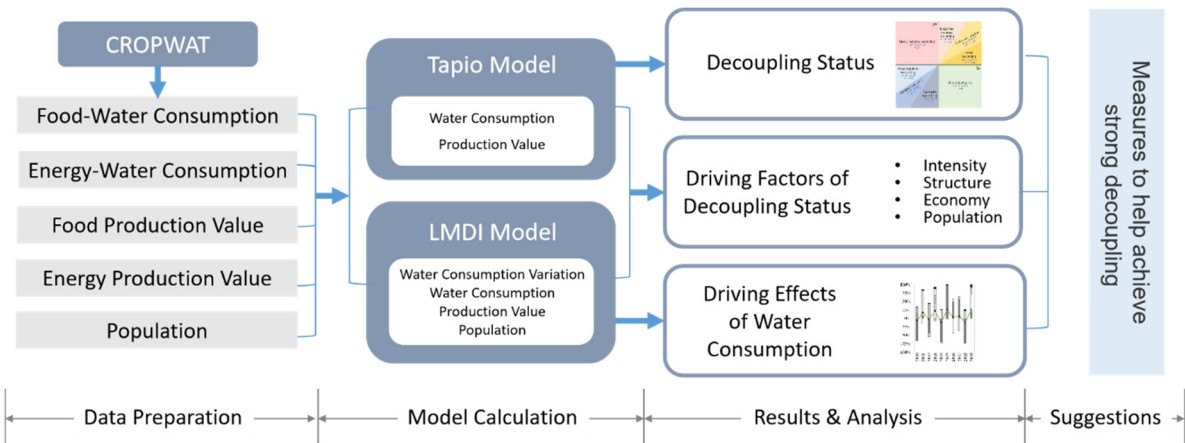

**Figure 2.** Analytical framework of driving factors of decoupling between economic development and water consumption in food and energy.

### 2.2.1. Data Sources

Relevant indicators and data sources are as follows: (1) The water demand of reference crops was calculated with CROPWAT 8.0 developed by FAO. When using the CROP-WAT model, the monthly average maximum temperature, monthly average minimum temperature, humidity, wind speed, and sunshine time from 2009 to 2019 were all from 22 meteorological stations in the Inner Mongolia Autonomous Region and Shaanxi Province in the China Meteorological Data Network (http://data.cma.cn, accessed on 26 March 2021). (2) Crop coefficient and crop growth period were referred to *Water Requirement and Irrigation of Main Crops in China* [34] and *Guideline for Computing Crop Water Requirement*, a series of irrigation documents published by FAO [35]. (3) Planting area, price index, energy production value, and population information was obtained from the *Inner Mongolia Statistical Yearbooks* and the *Shaanxi Statistical Yearbooks* from 2009 to 2019. (4) Food production value was considered as the product of food unit price and food planting area [36]. The food unit price was obtained by conversion of the food price index from 2010 to 2019, based on the *Compilation of China Agricultural Product Cost-Benefit Data* in 2009 [37].

### 2.2.2. Models and Methods

(1) Model of food water consumption

According to the *Inner Mongolia Statistical Yearbook*, the *Shaanxi Statistical Yearbook*, and the types of crops in the two provinces, wheat, corn, and soya account for the largest proportion of the planting area. In the Inner Mongolia Autonomous Region, the proportion of wheat, corn and soya in grain crops is 10%, 60%, and 20%, respectively, a total of 90%; in Shaanxi Province, the proportion of wheat, corn and soya in grain crops is 30%, 40%, 10%, respectively, a total of 80%. Wheat, corn, and soya are taken as accounting objects of food–water. The calculation formula of food–water consumption is as follows:

$$W_F = \sum_i W_{Fi} = W_{wheat} + W_{corn} + W_{soya} \tag{1}$$

where $W_F$ denotes the food–water consumption (m$^3$); $W_{Fi}$ is the water consumption of food $i$ (m$^3$); $W_{wheat}$ is the water consumption of wheat (m$^3$); $W_{corn}$ is the water consumption of corn (m$^3$); and $W_{soya}$ is the water consumption of soya (m$^3$).

The food water consumption calculation formula is recommended by FAO, and the food–water consumption at each growth stage is calculated by the food coefficient method.

$$W_{Fi} = ET_c \times S_i \tag{2}$$

$$ET_c = K_c \times ET_0 \tag{3}$$

In this expression, $S_i$ is the planting area of food $i$ (m$^2$); $ET_c$ refers to the evapotranspiration required for the growth of food under the assumption of optimal conditions, so the value is equal to the water consumption of food (mm); $K_c$ denotes the food coefficient (dimensionless); and $ET_0$ is the reference food–water consumption (mm), which was calculated with the CROPWAT model. The food growth stage and food coefficient ($K_c$) and the evapotranspiration required for the growth of food ($ET_c$) are shown in Appendix A.

(2) Model of energy–water consumption

According to the characteristics of energy structure, and the *Inner Mongolia Statistical Yearbooks*, and the *Shaanxi Statistical Yearbooks* in the two provinces, coal, oil, and natural gas together account for 97% and 98% of energy production in the two provinces, respectively. Because the production of clean energy uses almost no water. The water consumption of energy is mainly concentrated in coal, oil, and natural gas exploitation industries. The energy–water consumption calculation formula is as follows:

$$W_E = \sum_i W_{Ei} = W_{coal} + \left( W_{oil} + W_{gas} \right) \tag{4}$$

$$W_{Ei} = O_i \times w_i \tag{5}$$

where $W_E$ denotes the energy–water consumption (m$^3$); $W_{Ei}$ refers to the water consumption of energy $i$ (m$^3$); $W_{coal}$ is the coal–water consumption (m$^3$); $\left( W_{oil} + W_{gas} \right)$ is the water consumption of oil and natural gas (m$^3$); $O_i$ refers to the output of energy $i$ (t or m$^3$); and $w_i$ means the water consumption quota of energy $i$ (m$^3$/t or m$^3$/m$^3$), referring to the *Industrial Water Consumption Quota Standard of Inner Mongolia Autonomous Region(DB15/T385-2009)*, *Industrial Water Consumption Quota of Local Standard of Shaanxi Province (DB61/T943-2014)*, and Xiang's [38] research on energy and water consumption.

2.2.3. Decoupling Status between Economic Development and Water Consumption Based on the Tapio Model

Based on the research of Pan [39] and Li [40], this study used the Tapio model to analyze the decoupling relationship between production value and water consumption in food and energy in North-west China, and constructed the decoupling relationship model as follows:

$$T(W, V) = \frac{\Delta W / W}{\Delta V / V} \tag{6}$$

In this expression, $T(W, V)$ indicates the decoupling index between production and water consumption; $W$ denotes water consumption (m$^3$); $V$ denotes production value (USD); $\Delta W$ is the variation of water consumption (m$^3$); and $\Delta V$ is the variation of production value (USD).

Figure 3 shows the discriminant methods on the basis of Vehmas's [9] and Tapio's [10] research results. Table 1 illustrates the meaning of each decoupling status.

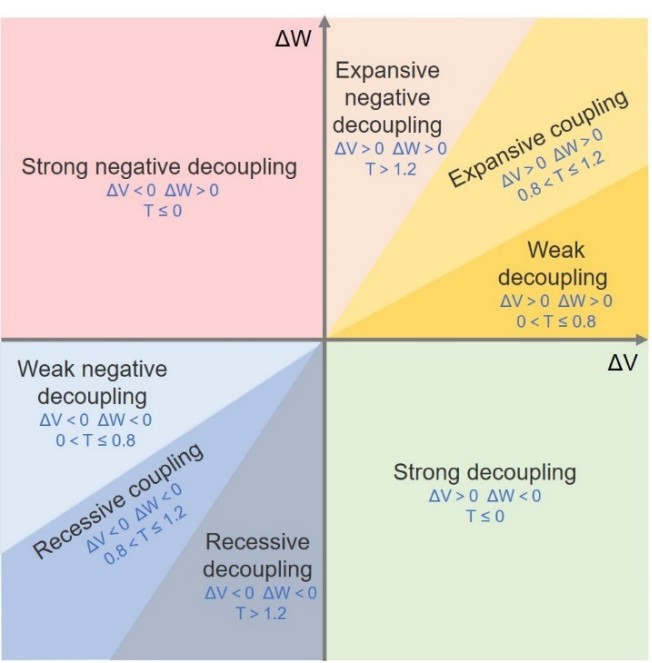

**Figure 3.** Decoupling classification and decoupling status discrimination.

2.2.4. Driving Factors of Decoupling Status Based on the LMDI Model

On the basis of using the Tapio model to evaluate the decoupling status, the driving effects of water consumption variation are divided into four effects: water intensity, structure, economy, and population by using the LMDI model. The formula for calculating the food (or energy) water consumption is as follows:

$$
\begin{aligned}
W &= \sum_i W_i \\
&= \sum_i \frac{W_i}{V_i} \times \frac{V_i}{V} \times \frac{V}{P} \times P \\
&= \sum_i I_i \times S_i \times Inc \times P
\end{aligned}
\tag{7}
$$

where $W_i$ is the water consumption of food $i$ (or energy $i$); $V_i$ is the production value of food $i$ (or energy $i$); $V$ is the total food (or energy) production value and $V = \sum_i V_i$; $P$ is the population of the region. $I_i = \frac{W_i}{V_i}$ denotes the water intensity for food (or energy); $S_i = \frac{V_i}{V}$ refers to the production value proportion of food $i$ (or energy $i$); and $Inc = \frac{V}{P}$ is the gross production value of food (or energy) per capita.

$\Delta W$ referring to the amount of water consumption changing with time can be decomposed into [25]:

$$
\Delta W = W^t - W^{t-1} = \Delta W_I + \Delta W_S + \Delta W_{Inc} + \Delta W_P
\tag{8}
$$

$$
\Delta W_I = \sum_i \frac{W_i^t - W_i^{t-1}}{\ln W_i^t - \ln W_i^{t-1}} \ln\left(\frac{I_i^t}{I_i^{t-1}}\right)
\tag{9}
$$

$$
\Delta W_s = \sum_i \frac{W_i^t - W_i^{t-1}}{\ln W_i^t - \ln W_i^{t-1}} \ln\left(\frac{S_i^t}{S_i^{t-1}}\right)
\tag{10}
$$

$$
\Delta W_{Inc} = \sum_i \frac{W_i^t - W_i^{t-1}}{\ln W_i^t - \ln W_i^{t-1}} \ln\left(\frac{Inc^t}{Inc^{t-1}}\right)
\tag{11}
$$

$$
\Delta W_P = \sum_i \frac{W_i^t - W_i^{t-1}}{\ln W_i^t - \ln W_i^{t-1}} \ln\left(\frac{P^t}{P^{t-1}}\right)
\tag{12}
$$

In this expression, $\Delta W_I, \Delta W_s, \Delta W_{Inc}, \Delta W_P$ are the water intensity effect, the structural effect, the economic effect, and the population effect, respectively, which affect the variation of water consumption.

Combining Formulas (8)–(12), Formula (6) can be rewritten as

$$
\begin{aligned}
T(W,V) &= \frac{\Delta W/W}{\Delta V/V} \\
&= \frac{(\Delta W_I + \Delta W_S + \Delta W_{Inc} + \Delta W_P)/W}{\Delta V/V} \\
&= \frac{V}{W\Delta V}\Delta W_I + \frac{V}{W\Delta V}\Delta W_S + \frac{V}{W\Delta V}\Delta W_{Inc} + \frac{V}{W\Delta V}\Delta W_P \\
&= T_I + T_S + T_{Inc} + T_P
\end{aligned}
\tag{13}
$$

where $T(W,V)$ indicates the decoupling index between production and water consumption (dimensionless); $T_I, T_S, T_{Inc}, T_P$ are the decoupling variation caused by the water intensity effect, the structural effect, the economic effect, and the population effect, respectively.

## 3. Results

### 3.1. Decoupling Status of Production Value and Water Consumption of Food and Energy in North-West China

The evolution of the water consumption and production value of food and energy in North-west China from 2009 to 2019 is shown in Figure 4. The food–water consumption in the Inner Mongolia Autonomous Region decreased first and then increased. The food production value and energy production value both rose first and then decreased from 2009 to 2014, and the energy–water consumption gradually increased. In Shaanxi Province, the food–water consumption decreased steadily, and the food production value performed steadily. The energy–water consumption maintained an upward trend on the whole, whose trend was highly consistent with the energy production value. In conclusion, from 2009 to 2019, the water consumption for food was relatively stable in North-west China, while the water consumption for energy grew rapidly. The food production value and energy production value also show different changing trends with time. The trends that show in Figure 4 make it imperative to explorer the decoupling status between production value and water consumption in this region, and in order to improve the production value and reduce water consumption at the same time, policies should be formulated according to the driving factors of changes in water consumption.

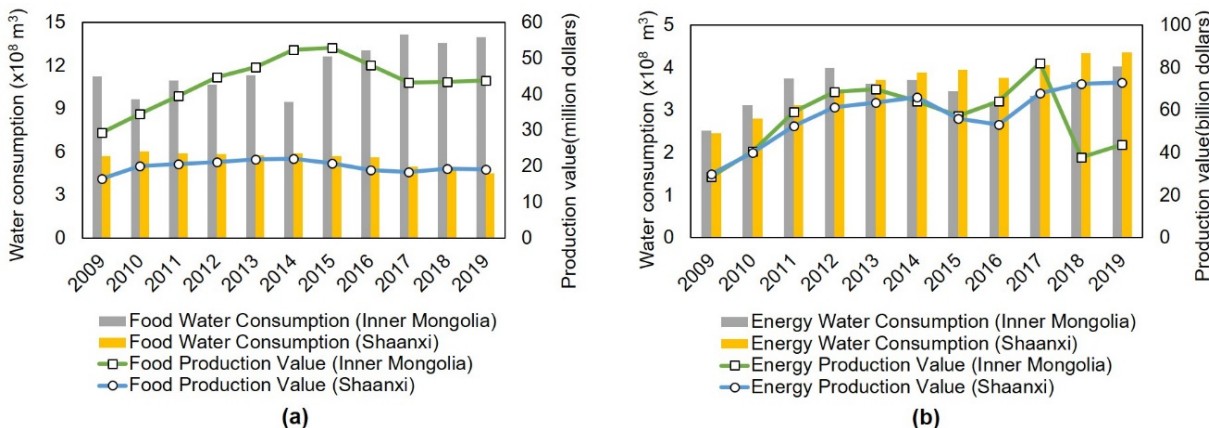

**Figure 4.** (**a**) Evolution of water consumption and production value of food in North-west China. (**b**) Evolution of water consumption and production value of energy in North-west China.

On the basis of Formula (6), the water consumption change rates, production value change rates, decoupling index, and decoupling status in the Inner Mongolia Autonomous Region (Table 2) and in Shaanxi Province (Table 3) from 2010 to 2019 were obtained.

**Table 2.** Decoupling index and decoupling status in the Inner Mongolia Autonomous Region.

| Water Consumption | Period | △V/V | △W/W | T | Decoupling Status |
|---|---|---|---|---|---|
| **Food** | 2010 | 0.18 | −0.14 | −0.78 | Strong decoupling |
| | 2011 | 0.15 | 0.13 | 0.91 | Expansive coupling |
| | 2012 | 0.13 | −0.02 | −0.19 | Strong decoupling |
| | 2013 | 0.06 | 0.06 | 0.92 | Expansive coupling |
| | 2014 | 0.1 | −0.16 | −1.62 | Strong decoupling |
| | 2015 | 0.01 | 0.33 | 29.68 | Expansive negative decoupling |
| | 2016 | −0.09 | 0.03 | −0.36 | Strong negative decoupling |
| | 2017 | −0.1 | 0.09 | −0.88 | Strong negative decoupling |
| | 2018 | 0 | −0.04 | −9.64 | Strong decoupling |
| | 2019 | 0.01 | 0.03 | 2.99 | Expansive negative decoupling |
| **Energy** | 2010 | 0.42 | 0.23 | 0.55 | Weak decoupling |
| | 2011 | 0.46 | 0.2 | 0.44 | Weak decoupling |
| | 2012 | 0.16 | 0.07 | 0.43 | Weak decoupling |
| | 2013 | 0.02 | −0.09 | −5.9 | Strong decoupling |
| | 2014 | −0.08 | 0.03 | −0.32 | Strong negative decoupling |
| | 2015 | −0.11 | −0.08 | 0.7 | Weak negative decoupling |
| | 2016 | 0.12 | −0.07 | −0.58 | Strong decoupling |
| | 2017 | 0.28 | 0.04 | 0.15 | Weak decoupling |
| | 2018 | −0.54 | 0.1 | −0.19 | Strong negative decoupling |
| | 2019 | 0.16 | 0.1 | 0.61 | Weak decoupling |

**Table 3.** Decoupling index and decoupling status in Shaanxi Province.

| Water Consumption | Period | △V/V | △W/W | T | Decoupling Status |
|---|---|---|---|---|---|
| **Food** | 2010 | 0.21 | 0.05 | 0.25 | Weak decoupling |
| | 2011 | 0.03 | −0.02 | −0.53 | Strong decoupling |
| | 2012 | 0.02 | −0.01 | −0.41 | Strong decoupling |
| | 2013 | 0.03 | −0.02 | −0.68 | Strong decoupling |
| | 2014 | 0.01 | 0.03 | 3.26 | Expansive negative decoupling |
| | 2015 | −0.06 | −0.03 | 0.56 | Weak negative decoupling |
| | 2016 | −0.09 | −0.01 | 0.15 | Weak negative decoupling |
| | 2017 | −0.03 | −0.12 | 3.84 | Recessive decoupling |
| | 2018 | 0.06 | −0.05 | −0.93 | Strong decoupling |
| | 2019 | −0.01 | −0.04 | 3.85 | Recessive decoupling |
| **Energy** | 2010 | 0.33 | 0.14 | 0.42 | Weak decoupling |
| | 2011 | 0.31 | 0.11 | 0.36 | Weak decoupling |
| | 2012 | 0.17 | 0.12 | 0.74 | Weak decoupling |
| | 2013 | 0.03 | 0.06 | 1.96 | Expansive negative decoupling |
| | 2014 | 0.04 | 0.04 | 1.13 | Expansive coupling |
| | 2015 | −0.15 | 0.02 | −0.13 | Strong negative decoupling |
| | 2016 | −0.05 | −0.05 | 0.9 | Recessive coupling |
| | 2017 | 0.28 | 0.08 | 0.27 | Weak decoupling |
| | 2018 | 0.06 | 0.07 | 1.14 | Expansive coupling |
| | 2019 | 0.01 | 0 | 0.19 | Weak decoupling |

In the Inner Mongolia Autonomous Region (Table 2), decoupling status between water consumption and production value in food displayed four situations: Strong decoupling, expansive coupling, expansive negative decoupling, and strong negative decoupling. The growth rate of the food production value gradually slowed down, and there was even a strong negative decoupling status of an extreme non-ideal situation during the period of 2015~2017. In 2018, the food–water consumption returned to strong decoupling status. The decoupling status of energy–water consumption in the Inner Mongolia Autonomous Region presents a changing process of "weak–strong–weak". In recent years, the production value and water consumption in energy were in a strong decoupling status. This indicates that the decoupling status between economic development and water resource

consumption in the Inner Mongolia Autonomous Region is unstable. In order to make the decoupling status stable in strong decoupling, it is necessary to further explore the factors affecting decoupling and then take corresponding measures.

In Shaanxi Province (Table 3), the strong decoupling status between production and water consumption in food mainly appeared before 2014. From 2011 to 2019, the rate of change in food production value stayed between −0.1 and 0.1, and began to decrease slightly in 2015, while the rate of food–water consumption fluctuated greatly. Except for the short-term strong decoupling in 2018, the decoupling status between production and water consumption in food from 2014 to 2019 mainly showed weak negative decoupling and recessive decoupling status. From 2010 to 2016, production value and water consumption in energy were of strong negative decoupling and a recessive coupling status; from 2017 to 2019, the production value and water consumption in energy showed a primary sustainable scenario trend with weak decoupling and expansive coupling status. Water is used excessively and inefficiently for food and energy production. It means too much water is being used for food and energy, and large amounts of water use do not always lead to higher food and energy production, which is not conducive to the sustainable development of Shaanxi Province. It is necessary to study the driving effect of water resources change and the driving factors of decoupling change to improve the decoupling status.

### 3.2. Driving Effects on Food and Energy–Water Consumption Variation in North-West China

On the basis of Formulas (7)–(12), driving effects on the water consumption variation were obtained, as shown in Figure 5. The contribution rates of the driving effect are shown in Figure 6.

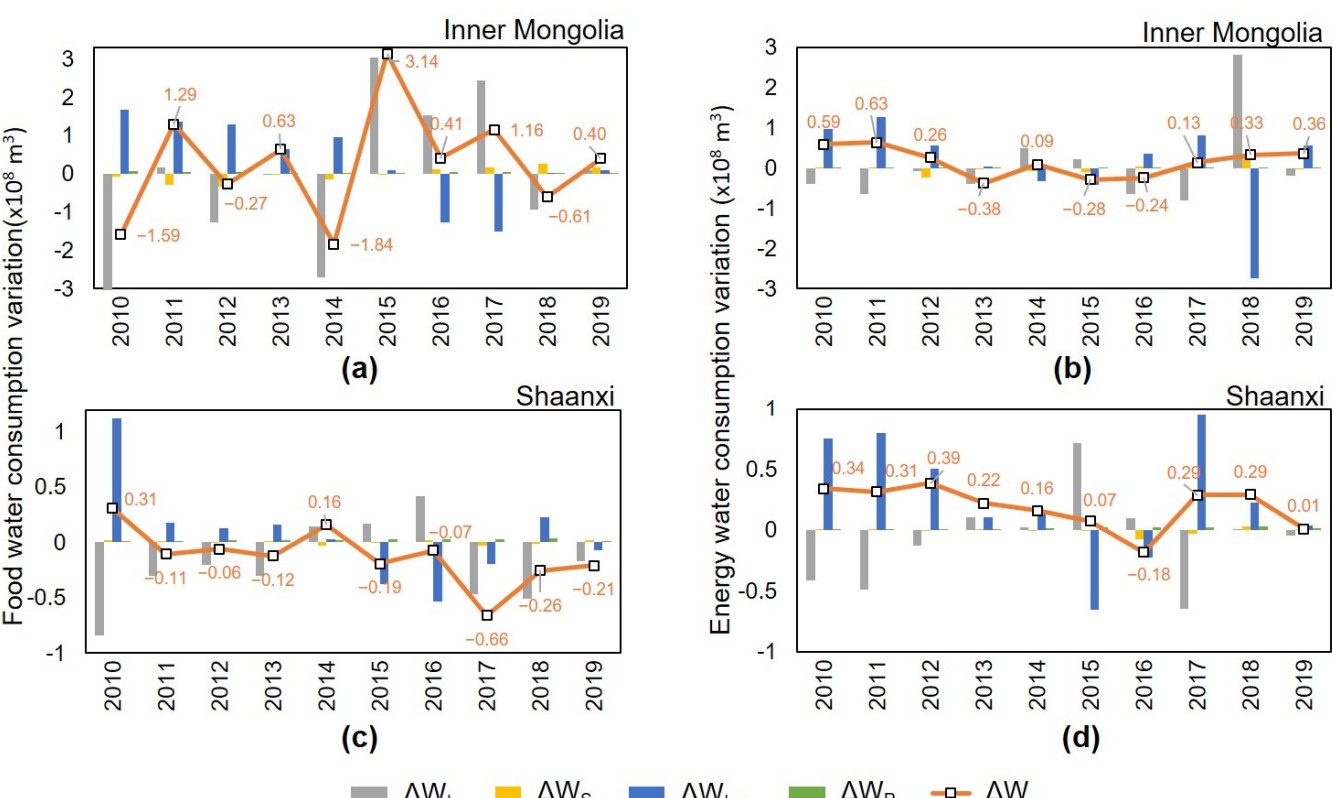

**Figure 5.** (**a**) Driving effect on food–water consumption variation in Inner Mongolia. (**b**) Driving effect on energy–water consumption variation in Inner Mongolia. (**c**) Driving effect on food–water consumption variation in Shaanxi. (**d**) Driving effect on energy–water consumption variation in Shaanxi.

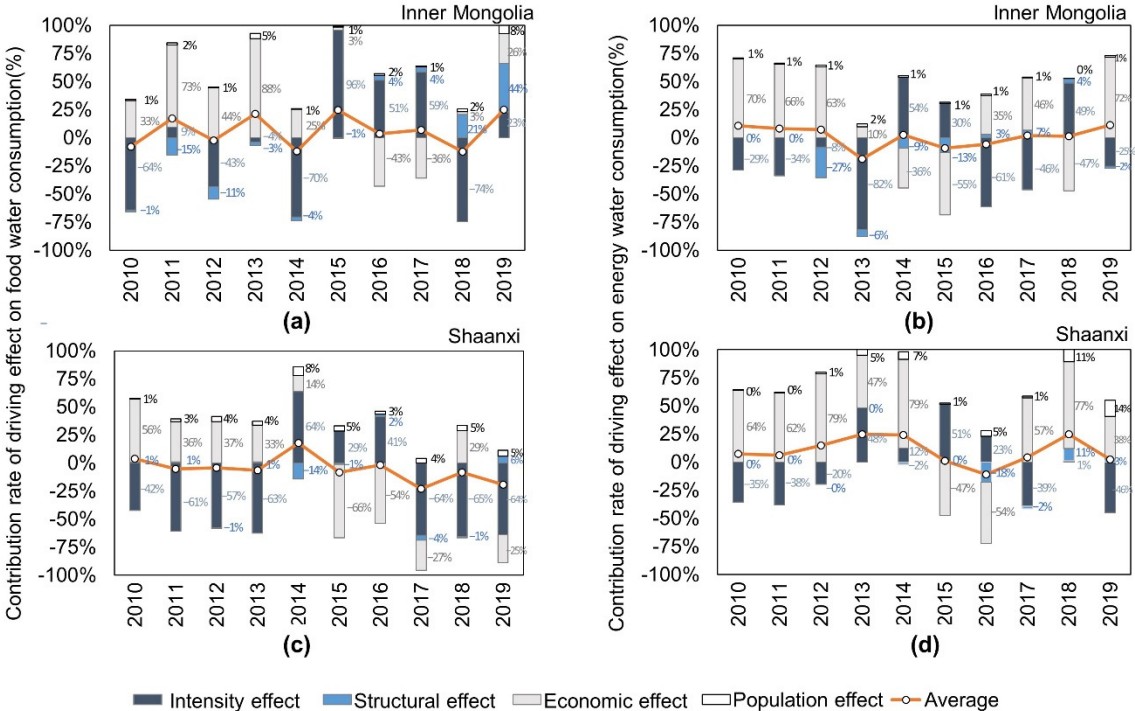

**Figure 6.** (**a**) Contribution rate of driving effect on food–water consumption in Inner Mongolia. (**b**) Contribution rate of driving effect on energy–water consumption in Inner Mongolia. (**c**) Contribution rate of driving effect on food–water consumption in Shaanxi. (**d**) Contribution rate of driving effect on energy–water consumption in Shaanxi.

It can be seen in Figure 5a that the food–water consumption in the Inner Mongolia Autonomous Region increased by $2.73 \times 10^8$ m$^3$ from 2009 to 2019. Figure 6a shows that the structural effect, economic effect, and population effect accounted for 37%, 216%, and 23% of the food–water consumption, respectively, which indicate these are the driving factors for the increase in food–water consumption, especially the economic growth; The intensity effect accounted for −19% of the food–water consumption, indicating that the decrease in water intensity was the driving factor for the reduction of food–water consumption. From the trend of each driving effect, the influence of water intensity on water consumption increases first and then decreases. The promoting effect of the structure adjustment on the growth of water consumption was gradually strengthened. Population growth has always promoted the growth of water consumption. Generally speaking, the intensity effect was the main driving factor of decline. Reducing the intensity of water use for food can promote the reduction of water consumption to a certain extent.

As can be seen from Figure 5b, the energy–water consumption in the Inner Mongolia Autonomous Region increased by $1.50 \times 10^8$ m$^3$ from 2009 to 2019. As presented in Figure 6b, the contributions of the economic effect and population effect to energy–water consumption were 223% and 11%, respectively, promoting the increase in water consumption together with the former being more significant. The proportions of the intensity effect and the structural effect on energy–water consumption were −152% and −44%, respectively. It can be inferred that the decrease in energy consumption would be mainly affected by them. Generally speaking, water consumption for energy has increased, which is positively correlated with the increased per capita energy production. The intensity effect is the main driving factor for energy water decrease, and the intensity of water use is directly related to trends in energy–water consumption.

As shown in Figure 5c, the food water consumption in Shaanxi Province decreased by $1.22 \times 10^8$ m$^3$ from 2010 to 2019. Figure 6c shows that the proportion of the intensity effect and the structural effect were −283% and −11%, respectively. Accordingly, they restrained the increase of water consumption, and the effect on the intensity effect was 25 times that of the structural effect. The economic effect and the population effect accounted for 34%

and 41% of food water consumption, respectively. The population effect played a more significant role in promoting the increase in water consumption. The decrease in food water intensity inhibits the increase in water consumption on the whole. After 2015, the decline in per capita food production began to affect the decline in food water consumption, and the contribution of population growth to the increase of food water consumption gradually weakened. In general, the decrease in food water intensity is the main driving factor for the decrease of water consumption, while the increase of population is the main driving factor for the increase of water consumption. Reducing water intensity and controlling population growth can adjust water consumption to a certain extent in Shaanxi Province.

From Figure 5d, it can be seen that from 2010 to 2019, energy and water consumption in Shaanxi Province increased by $1.90 \times 10^8$ m³. As presented in Figure 6d, the contributions of the economic effect and population effect to energy–water consumption was 402% and 46%, respectively, which caused an increase in energy and water consumption in Shaanxi Province, and the role of economic growth was more significant. The proportions of the intensity effect and the structural effect to energy–water consumption were −31% and −8%, respectively. It can be seen that they were the driving factors of reducing energy consumption. The decrease in water intensity for energy led to the decrease in energy–water consumption, but its contribution rate gradually weakened with the passage of time, and the energy–water consumption continued to increase. The increase in energy production per capita was continuously the main driving force for the increase in energy–water consumption, and its contribution rate was between 38 and 80%. A lower energy per capita production value can further promote the reduction of energy–water consumption.

*3.3. Driving Factors of Decoupling between Production Value and Water Consumption in Food and Energy in North-West China*

According to Formula (13), the changes in the decoupling driving factors were obtained as Table 4. The decomposition of the driving factors and their contribution rates are shown as Figure 7.

**Table 4.** Decoupling factor decomposition.

| Province | Period | Food | | | | | Energy | | | | |
|---|---|---|---|---|---|---|---|---|---|---|---|
| | | $T$ | $T_I$ | $T_S$ | $T_{Inc}$ | $T_P$ | $T$ | $T_I$ | $T_S$ | $T_{Inc}$ | $T_P$ |
| Inner Mongolia | 2010 | −0.78 | −1.60 | −0.03 | 0.82 | 0.03 | 0.55 | −0.38 | 0.00 | 0.91 | 0.02 |
| | 2011 | 0.91 | 0.12 | −0.20 | 0.97 | 0.03 | 0.44 | −0.46 | 0.00 | 0.90 | 0.01 |
| | 2012 | −0.19 | −0.89 | −0.23 | 0.90 | 0.02 | 0.43 | −0.12 | −0.40 | 0.94 | 0.02 |
| | 2013 | 0.92 | −0.04 | −0.03 | 0.95 | 0.05 | −5.90 | −6.36 | −0.49 | 0.76 | 0.18 |
| | 2014 | −1.62 | −2.37 | −0.12 | 0.85 | 0.03 | −0.32 | −1.66 | 0.28 | 1.09 | −0.04 |
| | 2015 | 29.68 | 28.70 | −0.18 | 0.89 | 0.26 | 0.70 | −0.57 | 0.25 | 1.04 | −0.02 |
| | 2016 | −0.36 | −1.32 | −0.11 | 1.11 | −0.04 | −0.58 | −1.56 | 0.07 | 0.88 | 0.03 |
| | 2017 | −0.88 | −1.84 | −0.13 | 1.13 | −0.03 | 0.15 | −0.88 | 0.13 | 0.89 | 0.01 |
| | 2018 | −9.64 | −14.69 | 4.08 | 0.51 | 0.47 | −0.19 | −1.56 | −0.13 | 1.51 | 0.00 |
| | 2019 | 2.99 | 0.67 | 1.31 | 0.78 | 0.23 | 0.61 | −0.34 | −0.02 | 0.96 | 0.01 |
| Shaanxi | 2010 | 0.25 | −0.69 | 0.01 | 0.92 | 0.01 | 0.42 | −0.51 | 0.00 | 0.91 | 0.01 |
| | 2011 | −0.53 | −1.53 | 0.02 | 0.91 | 0.06 | 0.36 | −0.56 | 0.00 | 0.91 | 0.01 |
| | 2012 | −0.41 | −1.36 | −0.03 | 0.88 | 0.11 | 0.74 | −0.24 | 0.00 | 0.96 | 0.02 |
| | 2013 | −0.68 | −1.68 | 0.03 | 0.88 | 0.10 | 1.96 | 0.95 | 0.00 | 0.92 | 0.09 |
| | 2014 | 3.26 | 2.90 | −0.65 | 0.66 | 0.35 | 1.13 | 0.14 | −0.02 | 0.92 | 0.08 |
| | 2015 | 0.56 | −0.47 | 0.02 | 1.09 | −0.08 | −0.13 | −1.23 | 0.01 | 1.13 | −0.03 |
| | 2016 | 0.15 | −0.85 | −0.04 | 1.10 | −0.06 | 0.90 | −0.47 | 0.36 | 1.10 | −0.10 |
| | 2017 | 3.84 | 2.71 | 0.18 | 1.13 | −0.18 | 0.27 | −0.61 | −0.03 | 0.89 | 0.02 |
| | 2018 | −0.93 | −1.84 | −0.04 | 0.82 | 0.13 | 1.14 | 0.01 | 0.12 | 0.88 | 0.12 |
| | 2019 | 3.85 | 3.18 | −0.31 | 1.25 | −0.26 | 0.19 | −0.86 | 0.05 | 0.72 | 0.27 |

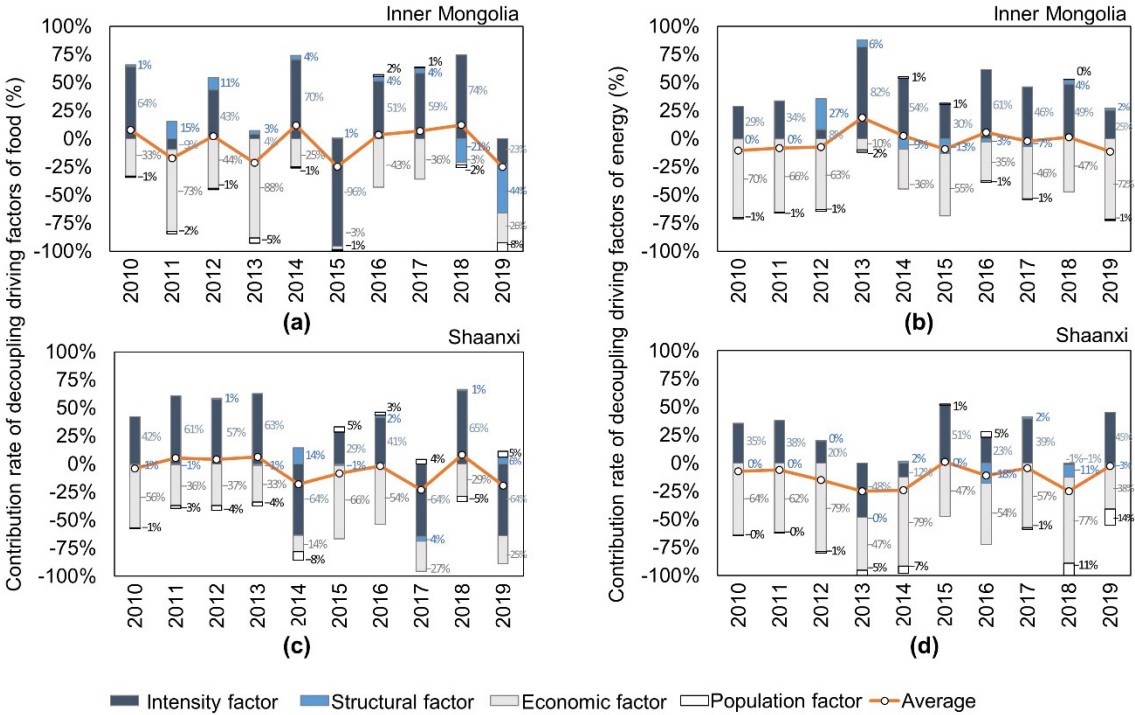

**Figure 7.** (**a**) Contribution rate of decoupling driving factors of food in Inner Mongolia. (**b**) Contribution rate of decoupling driving factors of energy in Inner Mongolia. (**c**) Contribution rate of decoupling driving factors of food in Shaanxi. (**d**) Contribution rate of decoupling driving factors of energy in Shaanxi.

From the perspective of decoupling between production value and water consumption in food in the Inner Mongolia Autonomous Region: Firstly, it can be seen from Table 4 that among the four driving factors, the main promoting factors of decoupling were the intensity factor and the structural factor, but the latter was weaker. The main restraining factor of decoupling was the economic factor. Secondly, Figure 7a shows that the overall contribution rate of the intensity factor was 239%, that of the economic factor was −375%, that of the structural factor was −20%, and that of the population factor was −18%. The promotion effect of the intensity factor on decoupling was consistent with the decoupling status of food production and water consumption in the Inner Mongolia Autonomous Region, that is, the reduction of food water intensity is conducive to the realization of decoupling.

From the perspective of decoupling between the production value and the water consumption in energy in the Inner Mongolia Autonomous Region: Firstly, it can be inferred from Table 4 that the decoupling index of energy production and water consumption in the Inner Mongolia Autonomous Region was positive except for the years 2013, 2014, 2016, and 2018. Among them, the intensity factor was the main promoting factor, while the economic factor was the main inhibiting factor. Secondly, it can be seen that from Figure 7b that the contribution rate of the intensity factor to decoupling was positive, and its overall contribution rate was 418%. The contribution rate of the economic factor to decoupling was negative, and its contribution rate was −499%. The contribution rates of the structural factor and the population factor were 7% and −6%, respectively. The promotion effect of the intensity factor and the inhibitory effect of the economic factor on decoupling were relatively stable. Reducing energy use intensity and increasing per capita energy production will help to achieve decoupling.

From the perspective of decoupling between production value and water consumption in food in Shaanxi Province: Firstly, Table 4 shows that the decoupling index between food production and water consumption in Shaanxi Province was positive except for in the years 2011, 2012, 2013, and 2019. Among four driving factors, the intensity factor was the main promoting factor while the economic factor was the main inhibiting factor. Secondly, Figure 7c shows that the overall contribution rate of the intensity factor was

116%, that of the economic factor was −376.58%, that of the structural factor was 17%, and that of the population factor was −6%. Except for in the years 2014, 2017, and 2019, water intensity factors promoted decoupling, and the contribution rate was more than 28%, which confirms that reducing water intensity for food is still important in realizing food water decoupling in Shaanxi Province. The restraining effect of the economic factors on decoupling also gradually weakened in the fluctuation, and the effect of the increasing per capita production value on the realization of decoupling decreased, while the population factor gradually began to play a promoting role on the realization of decoupling, indicating that a decrease in population is also gradually conducive to the realization of decoupling.

From the perspective of decoupling between the production value and water consumption in energy in Shaanxi Province: Firstly, as can be seen from Table 4, the decoupling index between energy production and water consumption in Shaanxi Province was positive, except for in 2015. Among the factors, the intensity factor was the main promoting factor, and the economic factor was the main restraining factor of decoupling energy production and water consumption in Shaanxi Province. Secondly, Figure 7d indicates that the overall contribution rate of the intensity factor was 190%, that of the economic factor was −604%, that of the structural factor was −28%, and that of the population factor was −34%. The intensity factor always promotes decoupling, the economic factor always restrains decoupling, and the structural factor gradually strengthens the restraining effect on decoupling. Therefore, it is necessary to pay attention to the change in energy exploitation structure while reducing water use intensity and increasing per capita production value.

## 4. Discussion

From the perspective of the "water–energy–food", this paper analyzed the decoupling status between production value and water consumption in food and energy in North-west China and explored the influencing factors of decoupling between production value and water consumption in food and energy from the perspective of the decoupling driving effect. The food production value and food water consumption fluctuated wildly, which shows that the decoupling status between economic development, and water consumption was diverse and that the influence relationship was complex. The discussion presented below is based on the above findings.

### 4.1. The Study Area Is Not Stable in the Ideal Status of Strong Decoupling

In the Inner Mongolia Autonomous Region, the reason for the strong negative decoupling of food–water consumption was the decline in corn price in the region, which accounted for about 90% of the total food output. In 2015, the national corn purchase price decreased by 11% year-on-year [41], resulting in lower food production value in the region, which caused the strong negative decoupling status of an extremely non-ideal situation. In 2018, the food production in the Inner Mongolia Autonomous Region reached a new high, surpassing that of the Sichuan Province and ranking first in the total food output in North-west China [42], and once again realized the strong decoupling. The decoupling status of energy production and water consumption in the Inner Mongolia Autonomous Region showed a "weak–strong–weak" change process. In 2013, the strong decoupling status that occurred in the Inner Mongolia Autonomous Region benefited from the formal implementation of the *Regulations of Inner Mongolia Autonomous Region on Saving Water* [43]. The strong negative decoupling status in 2018 was affected by the Inner Mongolia Autonomous Region's vigorous promotion of the coal de-capacity target in the *"13th Five-Year Plan"* [44], aims to close coal mining enterprises to less mining, reducing coal capacity. In addition, the growth trends of food–water and energy–water in the Inner Mongolia Autonomous Region are in the trade-off relationship, cycling so that when one falls, the other rises, indicating that energy and food compete for water. For all this, Inner Mongolia Autonomous Region has not yet reached an ideal status of strong decoupling, and the economic development of the in the region is still very dependent on the consumption of water resources.

In Shaanxi Province, after the strong decoupling status of food production and water consumption from 2011 to 2013, reduced arable land [45], and climate change [46] were increasingly restricting food production. In addition, water resources are in short supply, and the situation of grain and water decoupling is worsening. Re-coupling occurred in Shaanxi Province in recent years, and this province still faces the challenge of increasing food production values and water environment pressure [47]. The Shaanxi Province implemented the task of reducing coal by three million tons in 2015 [48], and a strict approval system for soil and water conservation programs was implemented due to the protecting ecology and controlling pollution action plan, which was released in 2016 [49]. This resulted in a decrease in energy output in 2016, which had a negative impact on the production value of energy. In short, in Shaanxi Province, the decoupling status of production and water consumption energy and food is not ideal.

### 4.2. Reduce Water Consumption by Adjusting the Intensity Effect

The economic effect had the greatest contribution to the variation of food and energy–water consumption in the Inner Mongolia Autonomous Region and energy–water consumption in Shaanxi Province. The intensity effect contributed to the reduction of food water consumption in Shaanxi Province. Generally speaking, the intensity effect was the main driving factor for the decline in water consumption, and the economic effect was the main driving factor for the increase in water consumption in North-west China. This is in agreement with Zhang's views that the main effect of production intensity restrains the increase in total water consumption [50], and the effect of regional economic development was the decisive factor for the rise in water consumption [51]. The authorities should realize that only by reducing water consumption will there be a chance for strong decoupling. Therefore, water resource policy needs to be performed simultaneously: The formulation of local energy and food industry planning should consider the regional water supply and demand, adopt economical water rationing to improve the relationship between water, energy, and food resources [52], so as to alleviate the contradictory relationship between them, and realize the maximum economic growth with minimum resource consumption. In addition, it is necessary to strengthen the protection of water sources and solve the problems of declining water sources and aging water source facilities in the region. Open-source water diversion, which means divert water from other areas to make up the shortfall, can be used as a primary measure to fundamentally solve water consumption difficulties [53]. At the ecological level, the government should also pay attention to the key technologies of sewage treatment, implement solutions to the problems existing in sewage discharge, and better integrate sewage recycling with ecological construction [54].

### 4.3. The Intensity and Structural Factors Promoting Decoupling

The intensity factor and the structural factor were the promoting factors of decoupling between production and water consumption in North-west China, and the former had a more remarkable effect. The economic factor and the population factor were the main restraining factors. The promotion effect of the intensity factor on food–water decoupling was less than that on energy–water consumption. The structural factor had a greater promoting effect on food water consumption than on energy–water consumption, and the promoting effect of the structural factor on food water decoupling in the Inner Mongolia Autonomous Region declined. The population factor had a positive effect on food water decoupling in Shaanxi Province. The observations regarding the importance of the intensity factor in promoting decoupling were similar to those of Ma [55], Yang [56], and Li [57]. Therefore, the authorities should pay attention to the influence of those driving factors on decoupling and implement targeted policies to strengthen decoupling: In terms of water use structure and intensity, it is necessary to optimize the industrial layout of energy and food, and to adjust the consumption structure. For energy–water, we should actively develop an advanced manufacturing technique, and improve the energy utilization efficiency [58,59], and for food–water, we can adopt water trading; that is, to reduce the

leakage of food–water in the transportation process through the lining of channels and the transformation of drip irrigation, and trade the surplus water resources to the industry in water-deficient areas [60]. We can also implement rotation of land cultivation [61], develop water-saving technologies, and construct irrigation facilities [62]. In addition, the negative contribution of the economic factor and the population factor to decoupling should be paid attention to [63]. The government should make a long-term plan to rationally control the population size, reduce the population pressure, keep the profitable industries closer to the concentrated population areas, actively implement various financial support, and promote labor enthusiasm so as to exchange more economic benefits with fewer labor costs.

*4.4. Limitations of the Research Framework*

It should be noted that there are still some limitations in this study: (1) In terms of the limitations of the model, the two provinces in North-west China selected as study area are distinctive in resource endowment and representative in economic construction, which leads to the analysis models being established based on the characteristics of the study areas. The dismantling of driving effects will vary in different space-time conditions, and this may affect the universality of the model in other areas. (2) In terms of the limitations of the analysis, the research on the decoupling of energy and food–water in this paper is independent, and the problem of multi-resource allocation caused by the possible competition between energy and food for water resources was not further explored. As the situation of energy–food competition for water is severe, to balance the allocation of water between energy and food production is a significant challenge facing researches as well as the government at present. Identifying how the flow relationship between water resources and food can be systematically handle in the future will be the main task in the later stage of this study.

**5. Conclusions**

There are some countermeasures and suggestions that are helpful in realizing the strong decoupling: (1) The protection of water sources for the supply of water resources should be strengthen, and attention should be paid to sewage treatment at the ecological level. (2) In terms of energy and food production, it is necessary to optimize the industrial layout, adjust the consumption structure, improve the efficiency of production and utilization, adhere to the principle of water resources recycling, and strengthen water-saving measures while improving production. (3) At the social level, the enthusiasm and efficiency of the labor force should be promoted through financial support.

In conclusion, water consumption data calculated by the CROPWAT model was innovatively used to support the accuracy of water resource data. In this study, the decoupling status between the economic development of food and energy and water resources consumption in North-west China have been analyzed, and the driving factors of decoupling have been deconstructed. This provides a new reference for alleviating the economic harm caused by the energy, food, and water crises in North-west China and provides a clear direction for formulating policy suggestions to realize the coordinated development of economic growth and water resources in the area. If developing countries want sustainable, stable, and rapid development, they can rely on the drivers sought in this article to formulate resource and development policies to ensure regional capacity for sustainable development.

**Author Contributions:** Conceptualization, Z.F. and J.H.; methodology, Z.P.; formal analysis, Z.P.; investigation, Z.P.; resources, J.H. and S.Y.; writing—original draft preparation, Z.P. and Z.F.; writing—review and editing, Z.F., J.C. and Y.X.; supervision, J.C. All authors have read and agreed to the published version of the manuscript.

**Funding:** This research was supported by the Suzhou University of Science and Technology Research Foundation (Humanities and Social Sciences) (XKR202008), Postgraduate Research & Practice Innovation Program of Jiangsu Province (B200203167) and The Fundamental Research Funds for the Central Universities (KYCX20_0509).

**Institutional Review Board Statement:** Not applicable.

**Informed Consent Statement:** Not applicable.

**Data Availability Statement:** Not applicable.

**Conflicts of Interest:** The authors declare no conflict of interest.

**Appendix A.**

*Appendix A.1. Food Growth Stage and Food Coefficient $K_c$*

FAO divides the growth period of foods into four stages: the early growth stage ($L_{ini}$), the rapid development period ($L_{dev}$), the mid-growth period ($L_{mid}$), and the and late growth stage ($L_{late}$). The food coefficients are $K_{cini}$ $K_{cmid}$ and $K_{cend}$. The water consumption of foods in each growth stage is calculated by the food coefficient method, referring to *Water Requirement and Irrigation of Main Foods in China* [34] to divide the growth periods, and the FAO *Guideline for Computing Crop Water Requirement* [35] to determine the food coefficient. The divided of growth stages of wheat, corn, and soya in North-west China and $K_c$ are shown as Table A1.

**Table A1.** Food growth stage and food coefficient.

| Food | Project | $L_{ini}$ | $L_{dev}$ | $L_{mid}$ | $L_{late}$ |
|---|---|---|---|---|---|
| Wheat | Month a | 4~5 | 5~6 | 6~8 | 8~9 |
| | Duration/a | 1 | 1 | 2 | 2 |
| | $K_c$ | 0.30 | — | 1.15 | 0.25 |
| Corn | Month a | 4~5 | 5~6 | 6~8 | 8~9 |
| | Duration/a | 1 | 1 | 2 | 2 |
| | $K_c$ | 0.30 | — | 1.15 | 0.40 |
| Soya | Month a | 4~5 | 5~6 | 6~8 | 8~9 |
| | Duration/a | 1 | 1 | 2 | 2 |
| | $K_c$ | 0.40 | — | 1.15 | 0.55 |

*Appendix A.2. The Evapotranspiration Required for the Growth of Food $ET_c$*

$ET_c$ refers the evapotranspiration required for the growth of food under the assumption of optimal conditions, so the value is equal to the water consumption of food (whose calculation formula is recommended by FAO), and the food water consumption at each growth stage is calculated by the food coefficient method. The calculation results are shown in Table A2.

**Table A2.** The evapotranspiration required for the growth of food.

| Province | Period | $ET_c$ (mm) | | |
|---|---|---|---|---|
| | | Wheat | Corn | Soya |
| Inner Mongolia | 2009 | 24.31 | 25.72 | 27.51 |
| | 2010 | 19.95 | 21.17 | 22.66 |
| | 2011 | 21.71 | 23.03 | 24.74 |
| | 2012 | 20.68 | 21.81 | 23.37 |
| | 2013 | 20.63 | 21.94 | 23.53 |
| | 2014 | 16.70 | 17.69 | 18.98 |
| | 2015 | 21.62 | 22.81 | 24.35 |
| | 2016 | 21.88 | 23.16 | 24.84 |
| | 2017 | 24.02 | 25.33 | 27.05 |
| | 2018 | 22.55 | 23.81 | 25.54 |
| | 2019 | 22.92 | 24.21 | 25.96 |
| Shaanxi | 2009 | 21.76 | 22.82 | 24.31 |
| | 2010 | 22.60 | 23.72 | 25.22 |
| | 2011 | 22.48 | 23.63 | 25.27 |
| | 2012 | 22.42 | 23.65 | 25.35 |
| | 2013 | 22.64 | 23.94 | 25.71 |
| | 2014 | 24.18 | 25.43 | 27.10 |
| | 2015 | 23.36 | 24.73 | 26.55 |
| | 2016 | 21.81 | 23.17 | 24.98 |
| | 2017 | 20.69 | 21.85 | 23.36 |
| | 2018 | 19.63 | 20.91 | 22.60 |
| | 2019 | 18.81 | 20.00 | 21.60 |

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
