# Peer review of "Driving Factors of Decoupling between Economic Development and Water Consumption in Food and Energy in North-West China—Based on the Tapio-LMDI Method"

_water, doi:10.3390/w13070917_

Round 1

Reviewer 1 Report

This manuscript was evidently resubmitted hastily, because there are quite a number of formatting and other issues that need to be addressed.  I have identified some of them in the line-by-line comments below, but others will need to be addressed during copy editing.  My specific comments are as follows:   

L49. The rationale for stating that it is “imperative to determine the decoupling relationship” is not evident.  The existence of a new theory does not create conditions that are imperative.  It would be better to state that the new theory creates an opportunity to better understand the relationship between economic development and resource consumption, or something like this.

L57. I suggest that you end this sentence after the word situation.

L67. No need to capitalize the word on.  Should be …..; on…..

L78. ….research focuses….

L123. Here and throughout, it is more correct to write of Shaanxi Province (as you do in L176) rather than of the Shaanxi Province. 

L152. It would be more straightforward to describe the relationship between water and food in these provinces as one of inefficiency, rather than pressure.  The issue is that too much water is being used to produce a given amount of food.

L153. Is it between the development of production and the use of water or just production and the use of water?  I do not understand why “development” is stated here.

L161/162. By “energy water consumption,” do you mean water consumption to produce energy, or consumption of energy and water to produce food (see also Fig. 2)?  Please clarify.

L207. Insert the word “together” between the word gas and the word account.

L257. Simpler to state: …both rose first and then decreased from….

L264. This is a good place to use the word imperative.  The trends that you show in Figure 4 make it imperative to…….

L295. Food and energy production always depend on water consumption!  What you mean is that water use is excessive or that water is inefficiently used for food and energy production.

L301. Driving effects

L304. It is not readily apparent from Fig. 6a how you arrive at a number like 216.38%.  Could you add a sentence to indicate how this was done?  The figure is clear, but how does the reader look at those light gray bars to arrive at the number 216.38%.  Also, there is no need to include places to the right of the decimal point in giving these percentage numbers.  These decimal places are not significant.  If you state 216.38%, this implies that there is a real difference between 216.38% and 216.39%, which cannot be the case.  It would be sufficient to just state 216%, etc, when dealing with these calculated numbers.

Fig. 7. This figure appears six times (!) in the copy of the manuscript that I received for review, and there are also repeated paragraphs in the text.    

L363. Figure 7(a) shows that….

L371. Remove the word “in”

L410. Should just be “Discussion”  Also L417 discussion is based

L422. Caused, not cased

L431. Please add a little more explanation of coal de-capacity targets.  What does this mean?  Less mining and use of coal?  This is not clear.

L432/433. trade-off relationship, cycling so that when one falls, the other rises…..

L441. Still faces

L464. What is open-source water diversion?  Please explain.

L478. Those

L484. What is water right conversion?  Conversion from what to what?

L503. The authors selected, the paper did not select!  The same is true at L515.  The authors did the calculations, not the paper.

L514. The authors should consider deleting the first two paragraphs in this section, which are restatements of the findings.  The third paragraph, on the other hand, deals with conclusions—steps that should be taken based on the findings reported in the paper.  I think that the paragraph that begins at L492 belongs in the Conclusions section and not the Discussions section.  After all, it begins with the words “In conclusion.”  Please take a hard look at all the text beginning at L492 and organize it to put conclusions into Conclusions and avoid as much repetition and restatement as possible.  

Reviewer 2 Report

See attached review.

Author Response

This manuscript is a resubmission of an earlier submission. The following is a list of the peer review reports and author responses from that submission.

Round 1

Reviewer 1 Report

The manuscript deals with driving factors of decoupling between economic development and water consumption in food and energy in northwest China. Authors used the Tapio model to evaluate the decoupling status between economy and water resources of food and energy industry in Northwest China, and the LMDI model was used to analyze the driving effect on water consumption variation, and to reduce water resource consumption is helpful to achieve strong decoupling. The driving factors of decoupling between energy and food production value and water consumption in Northwest China are studied further, which provided decision-making basis for local governments to ensure regional sustainable development ability, and set a reference for solving global sustainable development problems. Overall, the concept of the paper is proper and important. Please check the upper script of the units, eg. line 146. What is the novelty of the approach presented in the study? What lessons should authorities  draw from this analysis? This should be discussed in the point concerning discussion of the results. Add references to all of the equation, in the case You are not the Author of the equations, eg. The region in which the inquiry was conducted, What's distinctive about it, What's typical, as compared to the rest of the region? Therefore, I propose to consider the possibility of completing the last point for discussion on possibilities and limitations of the use of such analysis. Besides, there is no discussion about possible limitations of using the proposed modelling. The main achievements of this study should be presented and underlined in the Conclusion.

Reviewer 2 Report

Dear Authors, The manuscript submitted discusses a very important topic "Decoupling" for allocating and devising better strategies to ensure sustainability. Though the manuscript is an interesting read, I have some suggestions, recommendations, feedbacks. Authors evaluated decoupling factors based on modeling outcomes. The models carry some sort of uncertainty, can be of any kind, structural, inherent, others. Did authors account the uncertainty? If not, it is important to consider as uncertainty quantification would make a big difference in decoupling classification. For figure 1: Secondary Y-axis, it would be good to convert the cost to USD unless there is a specific rationale. Authors mentioned about the three major crop classes; wheat, corn, and soya. Due to different Kc values and time to maturity it would be good to analyze the impact of decoupling of each crop than food as a big picture, which is an important information to the stake holders.

Reviewer 3 Report

This manuscript is well conceived and addresses interesting WEF issues, but there are several important problems that must be corrected.  Most deal with data presentation and the relationship between the text and the figures.  I am not an expert on the models used here, and so it is possible that some of my comments reflect my lack of detailed expertise, but many interested readers will also not be experts on the models.  Like me, they will want to be able to understand the manuscript and its WEF implications, and so it is important that the authors make the manuscript accessible to this body of readers.

Introduction.  This is a very long section with detailed information about decoupling theory and related approaches such as the “two-stage Tapio-LMDI method.” But very little is revealed about the fundamental theory.  What is measured and why is the decoupling approach for analysis of food and energy in northwest China?  Your arguments between L46 and 52 are not very convincing.  You state that because Carter put the decoupling theory forward and Cesano pointed out that it is important, it is “imperative” to apply it in northwest China.  Why?  You are clearly enthused, to the point of providing too much detailed introductory information, but you do not adequately convey the fundamental rationale for all of your work.  Some of this becomes clear later on, but you should make this explicit in the Introduction.

L123/141.  In this paragraph about northwest China, you write about issues that are troubled, important, in need of being clarified, great practical significance, Etc., but you provide no references to support any of these opinions.  This contrasts with the 27 earlier references about various aspects of the coupling method.  Please add references in this section to provide more balance and make it clear why it is so important to do the study.

L144/153.  On the basis of the statistics that you supply here, I do not understand the basic conflict between water and food in the two provinces.  I can see that there is proportionately more coal produced than food produced, but why is this a conflict?  And what is the evidence that water shortage is the issue?  Your rationale needs to be clearly explained, and you should cite references to support your arguments.

L158/166.  It would be helpful if the text here would correspond to the diagram given in Fig. 2.  For example, CROPWAT has one element listed in the text (food water consumption), but there are four elements in the Figure. The difference between “Water Consumption Output Value” from Tapio and “Water Consumption Output Value Population” from LMDI is not clear.  Please cite the statistical yearbooks (I think this is Ref. 5).  Also, what is the significance of the red-colored letters W, P, and V that sometimes appear in the figure? 

L177.  I think you mean that crop coefficient and crop growth period came from (not referred to) the publications listed. 

L187.  CROPWAT has been published and is well known, so I recommend deleting the details about the model in this section.  But you should still include important details about input data (below under L189/190 and L202/204).

L189/190.  How large?  Please state the proportion accounted for by these three crops.

L202/204.  Be more specific here, too.  How much water goes into these various industries?

The information given in this table is very helpful in addressing the issues I raised above under Introduction.  Table 1 helps the reader to understand what you mean by decoupling and what you hope to discover, but this information comes too late.  The reader should have this understanding after having completed reading the Introduction.

Fig. 4.  This figure is very clear, but the text that accompanies it is unclear, especially the use of “increased in fluctuation” and “consistent with the fluctuation trend.”  This appears to be an issue of English usage that should be corrected.

Table 2.  This table, which is a matrix of 9 columns X 20 rows, is much too complex and unintuitive to be of use in helping readers to understand decoupling status.  It is very important information for an Appendix, but here, you need to simplify things using diagrams and colors, so that the reader can easily follow your logic in the accompanying text (L262/281).  This is critical for understanding of your manuscript—you need to make it very easy for the reader to look at the data and see that your statements are correct.  Only specialists will want to dig down into at all of the data as shown here.    

Line 266.  Do you know what caused this extreme situation?

Section 3.2.  This section analyzes each of the four sub-figures in Fig. 6 and relates this information to the corresponding sub-figures in Fig. 5.  There are several important issues that should be clarified.  I am not sure if they are due to English usage, the fact that I am not an expert on these models, or some other misunderstanding, but I want to point them out.  One is that although the reader is referred to Fig. 5 to see a 271 million cubic meter increase in food water consumption in Inner Mongolia between 2009 and 2019 (L286), this number is not in Figure 5.  The figure has no absolute numbers—everything is relative to baselines that are not stated.  None of the other absolute numbers claimed to be in Figure 5 are there either.

Another issue has to do with the way the data are plotted in Figure 6 and the use of the word total.  Four different kinds of component effects are plotted year-by-year as stacked bars in the figure, and a total is given as a data point for each year. The text seems to use total in a different way—although a number is not given, the total is apparently the sum of the different component effects over the entire 10-year period, e.g., +37%, +216%, +23%, and -19%.  But I am not so sure about this, because for any given year, the total is clearly not the sum of the four individual component effects.  It is also clearly not the average of the four individual component effects, and for the first year (2010), it starts out in different places—about -10% in a, +10% in b, 0 in c, and 5% in d.  I understand the general trends here, but I do not understand the details.  I hope that simple changes in the figure and text will suffice to make all of this clear to the reader.   

L365.  Not Figure 1, which is the map at the beginning of the manuscript.

L374.  The figure that starts at L366 is very similar to Figure 6, and I have the same questions about it.  The text here refers to contribution rates (implying change over time) in panel a of this figure, but I do not see how, for example, a rate of 238% for the intensity factor is derived from the data in panel a.  The same issues apply to the other panels and analysis of them.

L430.  I agree that the short-term fluctuations are wild.  You have done a nice job in discussing a variety of factors influencing decoupling and some steps that might be taken to improve the situation.  But I wonder—are there steps that should be taken to avoid the wildness, that is to say, the extreme year on year changes?  Some of this seems to be due to rapid response to new government policies and market signals.  Do IM and Shaanxi just have to live with these factors, or is there something that could be done to avoid their negative effects?

L449/450.  I am not sure what is meant here.  Water grows, then energy, then water, then energy?  Also why is this grim?

L460/461. If the action plan was released in 2016, the effects should have been seen after 2016, not from 2015 to 2016.  Please explain.

Reviewer 4 Report

Technical review of the draft manuscript is attached.
